# Incorporation of Bayberry Tannin into a Locust Bean Gum/Carboxycellulose Nanocrystals/ZnO Coating: Properties and Its Application in Banana Preservation

**DOI:** 10.3390/polym15163364

**Published:** 2023-08-10

**Authors:** Wenrui Chi, Tingting Li, Na Wei, Zijing Pan, Lijuan Wang

**Affiliations:** Key Laboratory of Bio-Based Materials Science and Technology of Ministry of Education, Northeast Forestry University, 26th Hexing Road, Xiangfang District, Harbin 150040, China

**Keywords:** locust bean gum, bayberry tannins, coating, fruit preservation

## Abstract

The application of polysaccharide-based coatings to prolong the shelf-life of fruits has attracted increasing attention. This study aims to develop a fruit coating comprising locust bean gum/carboxycellulose nanocrystals/ZnO (LCZ) blended with bayberry tannins (BT). The results revealed a significant increase from 4.89% and 11.04% to 29.92% and 45.01% in the free radical scavenging rates of 2,2-diphenyl-1-picrylhydrazyl and 2,2′-azino-di-[3-ethylbenzthiazthiazoline sulfonate] with the percentage of BT increasing from 0% to 5%, respectively. At a 5% of BT, the antibacterial activity against both *E.coli* and *S. aureus* exceeded 90% while simultaneously achieving excellent UV shielding (transmittance of 380–200 nm ≤ 0.19%). After 3 days of storage, uncoated bananas showed signs of browning, and their titratable acid and vitamin C (Vc) contents decreased from 0.57% to 0.30% and from 7.37 mg/100 g to 4.77 mg/100 g, respectively. However, bananas coated with LCZ containing 3% BT not only exhibited a better appearance, but also possessed higher titratable acid (0.44%) and Vc content (5.31 mg/100 g). This study provides a sustainable and multifunctional coating for fruit preservation.

## 1. Introduction

Fruits are a rich source of essential nutrients including vitamins, dietary fiber, organic salts, and other vital compounds. They play a crucial role in maintaining a healthy daily diet for humans [1]. However, their perishability and susceptibility to spoilage due to microorganisms and oxidation after harvest result in nearly half of the global fruit production being wasted annually, while many people worldwide continue to suffer from hunger [2,3,4,5]. Therefore, the extension of fruit shelf life is crucial in mitigating the global food crisis and reducing food waste. Currently, low-temperature refrigeration [6], gas regulation [7], plastic packaging [8], and waxing [4] are commonly utilized in fruit preservation. Although low-temperature refrigeration and gas regulation can effectively prolong the storage period of fruits, their implementation requires specialized equipment and high energy consumption, while diverse fruit types may necessitate varying technical parameters [9]. The utilization of plastic packaging derived from petroleum contributes to environmental pollution due to the accumulation of non-biodegradable waste [10]. Conventional fruit waxing techniques commonly employed in commercial settings not only compromise the quality of the produce but also present challenges for removal and lead to some nutrient loss through peeling [11]. Therefore, there is an urgent need for a method of preserving fruits that is characterized by a low cost, wide applicability, environmental sustainability, effectiveness and ease of removal.

Cellulose nanofiber/nitrogen-functionalized carbon dot-coated tangerines and strawberries exhibited mold growth inhibition and an extended shelf life [12]. The ripening and softening of harvested bananas coated with carboxymethyl cellulose were observed to be significantly delayed, indicating a potential application for extending the shelf life of fresh ones [13]. The quality and nutritional characteristics of chitosan-coated loquat fruit were extended for up to one week under storage conditions of 7 °C and 95% relative humidity (RH) [14]. Arabic gum coating effectively extended the storage life of tomato fruits by inhibiting their respiration rate and ethylene production [15]. The storage period of bananas was extended by 40% at 20 °C and 52% RH through the application of a sucrose ester/rice starch coating [16]. Although those coatings obviously kept fruit freshness, there were still some drawbacks. For instance, the use of chitosan dissolved in an acetic acid solution may result in residual acids on the fruit surface after coating formation, which could potentially have a negative impact on consumers’ perception of the fruit’s appearance and odor [1]. Furthermore, low viscosity of those polysaccharide solutions leads to a low adhesion to the surface of fruits [11]. Therefore, it is necessary to seek a polysaccharide whose solution is highly sticky at a low concentration.

Locust bean gum (LBG) is an optimal alternative because of the high viscosity of the solution at low concentration, as well as a safe material that exhibits biocompatibility, non-teratogenicity, and biodegradability [17,18]. Although LBG is easily film-forming and adherent on fruit surfaces, the poor mechanical and barrier properties are unsatisfactory for the application. Those drawbacks can be reinforced by incorporating carboxycellulose nanocrystals (C-CNC) [19]. In our previous study [11], ZnO was incorporated to enhance the bacteriostatic properties of the C-CNC/LBG coating, which effectively extended the shelf life of strawberries, though its antioxidant activity was still poor. Tannins, plant-derived phenolic compounds rich in hydroxyl groups, have been recognized as safe by the FDA and exhibit exceptional antioxidant, antibacterial and UV-absorbing properties [9,20]. A tannin-rich edible coating has been demonstrated to extend the shelf life of fruit crop effectively, offering a promising alternative for post-harvest preservation [21]. To date, there is no report on the utilization of BT in LBG/C-CNC/ZnO coating for fruit preservation.

In this study, we investigated the impact of bayberry tannin on LBG/C-CNC/ZnO composite coatings using Fourier-transform infrared (FTIR) spectroscopy, thermogravimetric analysis (TGA), and scanning electron microscopy (SEM). The mechanical, barrier, antibacterial, and antioxidant properties were evaluated. Additionally, 3T-LCZ was employed to coat bananas and assess the vitamin C (Vc), titratable acid content and weight loss rate.

## 2. Materials and Methods

### 2.1. Materials

C-CNC (cellulose type I; length: 100~500 nm, diameter: 4~10 nm) was offered by Guilin Qihong Technology Co., Ltd. (Guilin, China). BT was acquired from Lining Technology new material Co., Ltd. (Linyi, China). LBG was acquired from Lavia Biotechnology Co., Ltd. (Xi’an, China). Glycerol was offered by Damao Chemical Co., Ltd. (Tianjin, China). ZnCl2 (AR) was bought from Tianda Chemical Reagent Factory (Tianjin, China). NaOH (AR) was provided by Tianjin Yongda Chemical Co., Ltd. (Tianjin, China). Bananas were bought from a local market (Harbin, China).

### 2.2. Methods and Equipment

#### 2.2.1. Preparation of ZnO Clusters

ZnO clusters were synthesized according to our previous work [11]. The ZnCl_2_ (0.1 mol/L) solution was gradually added to a 0.3 mol/L of NaOH solution (V(ZnCl_2_):V(NaOH) = 1:1) under continuous magnetic stirring. After a duration of 10 min, the resulting mixture was subjected to reaction at 60 °C for 480 min. Subsequently, the precipitate was separated via centrifugation, rinsed, and dried for subsequent utilization.

#### 2.2.2. Preparation of Composite Coatings

A total of 4.0 g of LBG was added into 100 mL of distilled water, while 1% C-CNC (*w/w* of LBG basis), 5% ZnO clusters (*w/w* of LBG basis), and BT were separately added into 100 mL of distilled water. Among them, the C-CNC and ZnO clusters were dispersed under sonication for 30 s before being mixed with the other solutions and stirred at 85 °C for 2 h. Subsequently, 35% glycerol (*w/w* of LBG basis) was incorporated and stirred for an additional 30 min. After defoaming via ultrasonication, the coating solution was poured into a mold and dried at 60 °C for 24 h. The resulting coatings were designated as 0T-LCZ, 1T-LCZ, 2T-LCZ, 3T-LCZ, 4T-LCZ and 5T-LCZ according to the addition of BT (0%, 1%, 2%, 3%, 4% and 5%, *w/w* of LBG basis). Coatings made solely from LBG and glycerol were designated as LBGF, while the LC coating was composed of LBG, 1% C-CNC, and 35% glycerol.

#### 2.2.3. Characterizations

The FTIR spectra of 0T-LCZ, 1T-LCZ, 3T-LCZ, and 5T-LCZ were collected using a Nicolet iN10 spectrometer (Thermo Fisher Scientific Inc., Branchburg, NJ, USA) within the wavenumber range of 4000~600 cm^−1^. The XRD patterns of the ZnO, LC, LBG, BT and composite coatings were obtained using an X-ray diffractometer (PANalitical B.V., Almelo, The Netherlands). The morphologies of the random composite coatings, as well as uncoated and 3T-LCZ-coated bananas, were observed using an Apreo S Hivac scanning electron microscope (Thermo Fisher Scientific Inc., Branchburg, NJ, USA). The thermostability of composite coatings (8~10 mg) were evaluated via a Q500 thermal gravimetric analyzer (TA Instruments—Water LLC, New Castle, DE, USA) under nitrogen atmosphere from 20 °C to 600 °C at the heating rate of 10 °C/min. The haze and light transmittance of composite coatings were measured using a haze mater (CS-700, CHNSpec, Hangzhou, China) and spectrophotometer (UV-2600, Shimadzu, Kyoto, Japan), respectively.

#### 2.2.4. Performance Measurements

The average thickness of the coatings was measured randomly at 10 points using an ID-C112XBS micrometer from Tokyo, Japan. After conditioning at 53% RH for 12 h, the rectangular coatings (1.5 cm × 8.0 cm) were subjected to tensile strength (TS) and elongation at break (EB) testing using an XLW-PC electronic intelligent tensile machine (Jinan, China), with a strain rate of 300 mm/min. Each sample was tested in five replicates. The oxygen permeability (OP) of the composite coatings at 0% RH was measured using an oxygen transmission rate tester (OX/230, Jinan, China). Meanwhile, the water vapor permeability (WVP) of these coatings was obtained using the gravimetric method. Coating discs with an area of 16.60 cm^2^ were securely sealed onto weighing bottles containing 23.0 g of anhydrous calcium chloride, and then conditioned at 25 °C and 75% RH. The moisture amount through the coatings was periodically measured by weighing the bottles, allowing for the calculation of WVP using Equation (1).
WVP = (*k* × *d*)/(Δ*P* × *s*)(1)
where *k* represents the rate of moisture gain per unit time (g/s), *d* denotes the average coating thickness (mm), Δ*P* indicates the driving force (1753.55 Pa), and *s* stands for the area of exposed coating surface (m^2^).

The antibacterial efficacy of the coatings was assessed using the plate count method. Specifically, 10.0 mL of bacterial suspension was cocultured with 200 mg of decontaminated broken coating in a constant temperature shaker incubator with 180 rpm/min at 37 °C for 2 h. Subsequently, 0.2 mL of the mixture was applied to nutrient agar plates and incubation for 12 h at 37 °C before recording colony counts. A control experiment without the addition of broken coating was also conducted. The antibacterial rate was calculated using Equation (2):Bacteriostatic rate = (C − X)/C × 100%(2)
where C represents the colony number of the blank and X represents the colony number of the sample being measured.

A total of 4.0 mg of 2,2-diphenyl-1-picrylhydrazyl (DPPH) was dissolved in a 100 mL solution of 95% ethanol to prepare the working solution. Then, 30.0 mg (dry weight) of coatings was dissolve completely in 20.0 mL of distilled water by shaking. The shake solution (2.0 mL) was combined with DPPH working solution (10.0 mL) and allowed to react for 15 min, followed by the measurement of the absorbance at 517 nm, which was recorded as A. A blank control consisting of 2.0 mL distilled water was used instead of the shaking solution, and its absorbance at 517 nm was measured as $A_{0}$. The DPPH free radical scavenging rate was evaluated using Equation (3):(3)$$\mathrm{Scavenging}\;\mathrm{rate}=(A_{0}-A)/A_{0}\;\times \;100\%$$
where A and $A_{0}$ denote the absorbance of the sample and control group, respectively.

A total of 20.3 mg of 2,2′-azino-di-[3-ethylbenzthiazthiazoline sulfonate] (ABTS) and 3.5 mg of K_2_S_2_O_8_ were dissolved separately in 5.0 mL of distilled water. The two solutions were then mixed and reacted for 12 h at room temperature in the dark to obtain the ABTS cationic radical mother solution. The working solution was obtained by diluting the mother solution with anhydrous ethanol until its absorbance at 734 nm was within a range of 0.75 ± 0.2. A total of 1.0 mL of coating solution was mixed with 10.0 mL of ABTS working solution, and the absorbance $A^{\prime }$ at 734 nm was measured after a reaction time of 6 min. The blank control was established by measuring the absorbance $A_{0}^{\prime }$ at 734 nm using 1.0 mL of distilled water instead of the shaking solution. ABTS free radical scavenging rate was calculated using Equation (4):(4)$$\mathrm{Scavenging}\;\mathrm{rate}=(A_{0}^{\prime }-A^{\prime })/A_{0}^{\prime }\;\times \;100\%$$
where $A^{\prime }$ and $A_{0}^{\prime }$ denote the absorbance of the sample and control group, respectively.

#### 2.2.5. Fruits Preservation Effect Measurement

The bananas were immersed in a NaClO solution of 5.04 g/L for 300 s, and then rinsed to remove surface residues. Subsequently, the washed bananas were dried and coated with a 3T-LCZ coating solution for 30 s, while the uncoated ones were immersed in distilled water for the same duration as the control. Finally, changes in appearance of all bananas during storage at room temperature (25 °C) were recorded and their mass measured periodically. The weight loss rate (*W*) was calculated using Equation (5):

(5)$$W=(m_{0}-m_{n})/m_{0}\;\times \;100\%$$
where $m_{0}$ (g) represents the original weight of the bananas; and $m_{n}$ (g) denotes the weight of the bananas after n days.

The Vc content in bananas was determined using a titrimetric method specified in GB5009.86-2016. Briefly, 20.0 g banana puree was placed into a 100 mL volumetric flask and diluted with a 20.0 g/L oxalic acid solution. Subsequently, 10 mL of the resulting filtrate was titrated with 2,6-dichloroindophenol solution (0.1 g/L) until it exhibited a pink coloration. The Vc content was calculated using Equation (6):(6)$$X=((V_{1}-V_{2})\;\times \;T\;\times \;A)/m\;\times \;100\%$$
where X (mg/100 g) is the content of Vc in the bananas; $V_{1}$ (mL) is the volume of 2,6-dichlorophenol solution consumed for titrating the sample; $V_{2}$ (mL) is the volume of 2,6-dichlorophenol solution consumed for titrating the blank sample; T (mg/mL) is the titration of 2,6-dichlorophenol solution; A is the dilution ratio; and m (g) represents the sample mass.

The titratable acid content in bananas was determined using the titrimetric method in GB/T12456-2008. Specifically, 20.0 g of banana puree was placed into a 100 mL volumetric flask, and filtered after settling with water. Then, 0.2 mL of a 1% phenolphthalein indicator was added to 20 mL of filtrate before being titrated with a solution of 0.1 mol/L NaOH until the appearance of pink. The content of titratable acid was calculated using Equation (7).

(7)$$\mathrm{TA}=(c\times \;(V_{1}^{\prime }-V_{2}^{\prime })\;\times \;K\;\times \;F)/m\;\times \;100\%$$
where TA (%) represents the titratable acid content of bananas; c (mol/L) denotes the concentration of NaOH; $V_{1}^{\prime }$ (mL) indicates the volume of NaOH solution consumed by the test solution; $V_{2}^{\prime }$ (mL) refers to the volume of NaOH solution consumed by the blank; K is defined as the acid conversion factor, which is 0.067; F is the dilution ratio; and m (g) signifies the mass of the sample.

#### 2.2.6. Statistics Analysis

The data analysis was performed using SPSS v17.0 (SPSS Inc., Chicago, IL, USA) and the Duncan multiple range test (*p* < 0.05) was employed.

## 3. Results and Discussion

### 3.1. FTIR Analysis

Figure 1 displays the FTIR spectra of LBG, C-CNC, BT, ZnO and composite coatings. Bands at approximately 3337 cm^−1^, 2929 cm^−1^, 2888 cm^−1^, 1653 cm^−1^, 1017 cm^−1^ and 868 cm^−1^ are associated with the O-H stretching vibrations, C-H stretching vibrations, C-H bending vibrations, aldehyde group at LBG chain ends, C-O stretching vibrations of pyranose and C-O-C stretching vibrations of glycosidic bonds, respectively [22,23,24]. The BT spectrum exhibits characteristic bands at 1608 cm^−1^ (C=C) and 1447 cm^−1^ (C-H) for the benzene skeleton [25]. The band detected at 1342 cm^−1^ is contributed by aryl hydroxyl groups [26]. Additionally, the band observed at 885 cm^−1^ is indicative of ZnO [27,28]. Those bands were not significantly different in the spectra of the composite coatings, indicating that no chemical reactions occurred among the components (An enlarged figure of the 0T-LCZ and 5T-LCZ is presented in Appendix A).

### 3.2. XRD Analysis

Figure 2 illustrates the XRD patterns of LBG, LC, BT, ZnO and the composite coatings. The characteristic peaks of LBG are observed at 2θ of 5.6°, 10.9°, 16.6°, 20.1° and 23.33°. After blending with C-CNC, the XRD pattern of LC showed no obvious changes. For ZnO, characteristic peaks at 31.8°, 34.4°, 36.3°, 47.6°, 56.6°, 62.9° and 67.9°, attributed to the (100), (002), (101), (102), (110), (103) and (112) crystal planes of the hexagonal wurtzite [29,30]. BT displayed an amorphous structure without any diffraction peak. The XRD pattern of 0T-LCZ showed typical ZnO peaks, indicating that there was no alteration in the crystal structure of ZnO during the coating production. Furthermore, the addition of BT did not cause any effect on the structure of other substances in the composite coatings.

### 3.3. Thermal Stability 

The thermal stability of the composite coating and BT is illustrated in Figure 3, which demonstrates four distinct weight loss processes. The first stage involved the evaporation of water from room temperature to 120 °C, followed by the second stage (120~255 °C) which resulted from glycerol volatilization and BT decomposition. At this stage, hydrogen bonding in the composite coating was increased by the incorporation of BT, resulting in a reduction in the thermal decomposition rate. Thirdly, in the range of 255~345 °C, thermal degradation of the LBG and C-CNC molecular chains occurs. The fastest rate of thermal decomposition for coatings decreased from 298.5 °C to 295.5 °C as the addition of BT increased from 0% to 5%. This indicated that the incorporation of BT effectively impeded the formation of hydrogen bonds among polymers. The final stage of mass loss (345~520 °C) corresponded to the subsequent thermolysis process involving pyrolysis products.

### 3.4. Physical Properties

The mechanical properties, OP, and WVP of the composite coatings are presented in Table 1. With increasing BT content, the TS of the coating initially decreased, and then increased before decreasing again; however, there was no significant change in EB. The TS increased from 31.80 MPa to 44.57 MPa upon the addition of 3% BT, and subsequently decreased to 39.17 MPa with the addition 5% BT. This could be attributed to the reduction in original hydrogen bonding among LBG, C-CNC and ZnO caused by BT, while new hydrogen bonds were formed with those components. With smaller BT additions, the combination of those two forces resulted in a reduction in TS, while larger additions led to a higher one. However, excessive addition caused mechanical property reduction, which could be attributed to the reduction in hydrogen bonds between the polymers. As the BT content increases, the barrier (oxygen and water vapor) of the coatings followed a similar trend to TS. At a 3% BT addition, the OP and WVP decreased from 0.72 cm^3^ mm m^−^^2^ day^−^^1^ atm^−^^1^ and 1.90 × 10^−^^11^ g m^−^^1^ s^−^^1^ Pa^−^^1^ to 0.66 cm^3^ mm m^−^^2^ day^−^^1^ atm^−^^1^ and 1.67 × 10^−^^11^ g m^−^^1^ s^−^^1^ Pa^−^^1^, respectively. Conversely, at 5% addition, the OP and WVP increased to 0.92 cm^3^ mm m^−^^2^ day^−^^1^ atm^−^^1^ and 1.71 × 10^−^^11^ g m^−^^1^ s^−^^1^ Pa^−^^1^, respectively. The reason for this was similar to the change in TS. Initially, the addition of a small amount of BT disrupted the intermolecular bonds among LBG, C-CNC and ZnO, resulting in an increase in OP and WVP. However, with further addition, new intermolecular hydrogen bonds were formed among the components, creating a dense network structure once again. As a result, water vapor molecules encountered difficulty in the penetration, while oxygen molecules followed a more convoluted path. With increasing amounts of additive, the dense structure was disrupted, resulting in higher OP and WVP. Additionally, excessive incorporation of hydroxyl-rich BT also increased the rate of water vapor transmission.

### 3.5. Optical Performance and Antibacterial Activity

The optical properties of the composite coatings are presented in Figure 4a,b. With an increase in BT addition from 0% to 5%, the light transmittance of the coatings decreased from 9.62% to 5.63%, while the haze increased from 82.71% to 89.89%. Those changes could be attributed to BT causing a more disordered structure of the coating, resulting in enhanced light scattering and reflection. Furthermore, the UV shielding capability of the composite coatings exhibited an increasing trend with the BT content. The transmittance of 5T-LCZ did not exceed 0.19% within the wavelength range of 200–380 nm. This phenomenon could be caused by the presence of a benzene ring structure in BT, which possesses a π–electron cloud capable of absorbing UV radiation and generate electron leaps [20]. Furthermore, as depicted in Figure 4c, the coating’s color shifts from a pale yellow to a claybank shade with BT content. Therefore, it was crucial to select an appropriate amount of BT addition when applying to fruit-coated coatings. As shown in Figure 4d, the coatings’ antibacterial activity against E. coli and S. aureus increased significantly with the addition of BT from 1% to 5%, from 4.5% to 92.59% and 32.54% to 94.61%, respectively. The abundant phenolic hydroxyl groups in BT could disrupt the integrity of bacterial cell membranes and walls, resulting in the leakage of cellular components and thus exhibiting bactericidal activity [31]. The potent antibacterial effect is crucial for inhibiting microbial growth and reducing fruit spoilage rates.

### 3.6. Antioxidation Property

Figure 5 shows the antioxidative performance of the coatings, revealing that the addition of BT from 0% to 5% in the composite coatings resulted in an increase in scavenging rates for DPPH• and ABTS• from 4.89% to 29.92% and from 11.04% to 45.01%, respectively. Coatings with excellent antioxidant properties could effectively eliminate the free radicals generated during fruits storage, thereby inhibiting fruit browning. Consequently, those composite coatings could extended the shelf life of fruits.

### 3.7. SEM

Figure 6 illustrates the surface and cross-section morphology of the composite coating, as well as a comparison between uncoated and coated 3T-LCZ bananas. The incorporation of a small quantity BT resulted in a smooth coating surface, while exceeding 3% led to an increase in roughness on the surface, which may account for the deterioration in mechanical properties beyond this threshold. Nevertheless, the coating remained dense and uninterrupted, exhibiting no voids or gaps. The uncoated banana surface exhibited a pronounced roughness with visible pores, whereas the coated banana displayed a smooth surface with covered pores. This observation implied that the coating may effectively reduce the respiration intensity of bananas. Furthermore, the cross-section of the banana revealed a typical porous structure, and it is evident that the 3T-LCZ coating was tightly adhered to the banana surface.

### 3.8. The Preservation Effect of Coatings on Bananas

In combination with those properties, 3T-LCZ was applied to coat bananas and assess its effectiveness in preservation. As depicted in Figure 7, the uncoated banana turned completely brown after a span of 9 days. This phenomenon could be attributed to the chemical reaction between polyphenol oxidase and oxygen, resulting in the production of melanin through enzymatic browning. Consequently, the flesh of the banana darkened and became unsuitable for consumption. Bananas coated with 3T-LCZ exhibited slight browning on the skin and completely darkening of the stems, while the flesh remained fresh. This was attributed to the excellent oxygen barrier and antioxidant properties of 3T-LCZ, which efficiently reduced enzymatic browning reaction. SEM observations revealed that 3T-LCZ tightly adhered the banana surface and effectively covered stomata. Meanwhile, the weight loss of uncoated bananas reached 29.18% after being stored for 9 days. However, bananas coated with 3T-LCZ exhibited a remarkable reduction in weight loss by up to 45.03%, which was attributed to the effective inhibition of respiration and transpiration processes through the coating treatment. Furthermore, it was worth noting that significant browning occurred on uncoated bananas after only 3 days of storage. As shown in Table 2, their titratable acid and Vc contents decreased from 0.57% to 0.30% and from 7.37 mg/100 g to 4.77 mg/100 g, respectively. However, 3T-LCZ-coated bananas maintained their titratable acid and Vc content at 0.44% and 5.31 mg/100 g, respectively. Those results demonstrated the effectiveness of 3T-LCZ in extending the shelf life of bananas.

## 4. Conclusions

LBG/C-CNC/ZnO/BT coatings, suitable for fruit preservation, were successfully prepared. The incorporation of 3% BT resulted in an increased TS of 44.57 MPa and decreased OP and WVP to 0.66 cm^3^ mm m^−^^2^ day^−^^1^ atm^−^^1^ and 1.67 × 10^−^^11^ g m^−^^1^ s^−^^1^ Pa^−^^1^, respectively. The gradual addition of BT enhanced UV resistance, and the antioxidant and bacteriostatic properties of the coatings. This provided a theoretical basis for its potential application as a fruit preservation coating. Moreover, when 3T-LCZ was applied to bananas, the coating tightly adhered to the surface and covered the air holes. The results indicated that LBG/C-CNC/ZnO/BT coatings effectively suppressed respiration, prevented browning, reduced weight loss rate, and reduced nutrient loss in bananas. This multifunctional coating was low-cost and sustainable, offering a novel method for fruit preservation and promising to alleviate the global food waste problem.

## Figures and Tables

**Figure 1 polymers-15-03364-f001:**
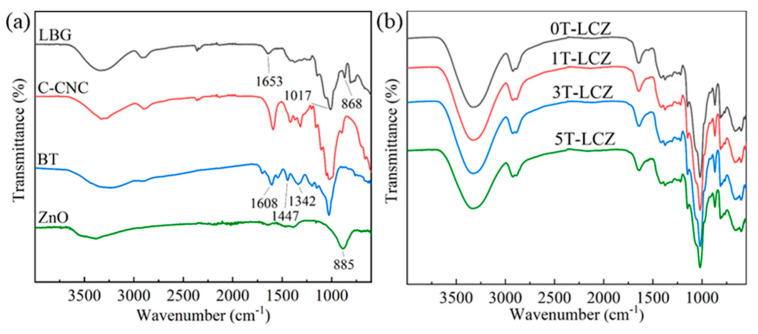
FTIR spectra of matrix materials (**a**) and the composite coatings (**b**).

**Figure 2 polymers-15-03364-f002:**
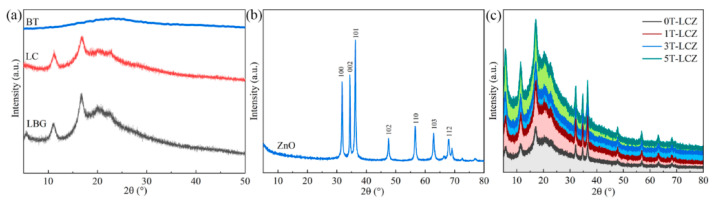
XRD patterns of LBG, LC and BT (**a**); ZnO (**b**) and the composite coatings (**c**).

**Figure 3 polymers-15-03364-f003:**
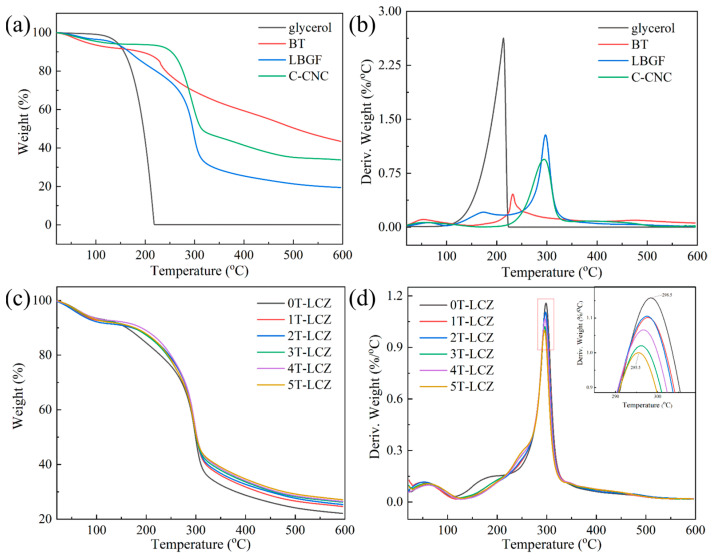
TG (**a**,**c**) and DTG (**b**,**d**) curves of glycerol, BT, LBGF, C-CNC and composite coatings.

**Figure 4 polymers-15-03364-f004:**
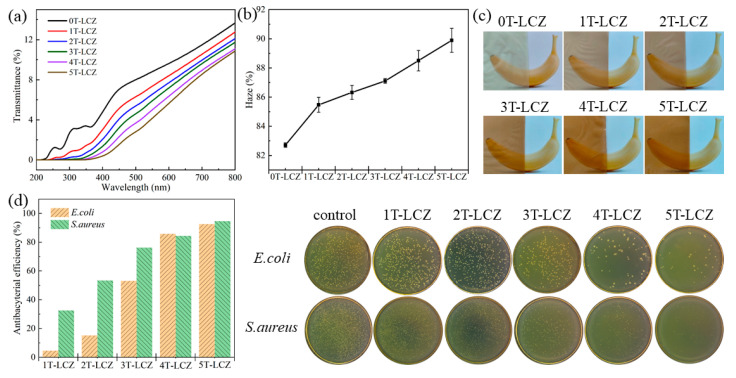
Transmittance (**a**), haze (**b**) of composite coatings, bananas’ pictures of coating cover (**c**) and antibacterial activity of the composite coatings (**d**).

**Figure 5 polymers-15-03364-f005:**
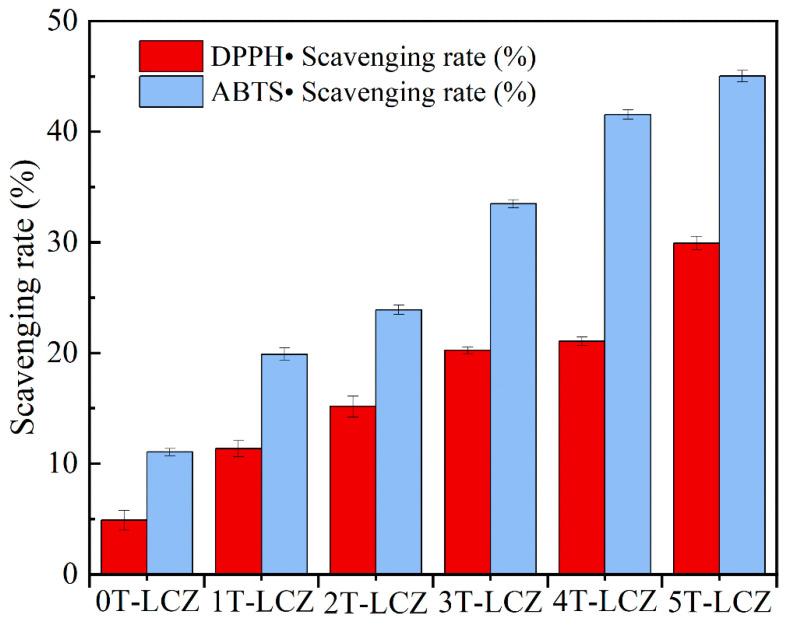
Scavenging rate of composite coatings against DPPH• and ABTS•.

**Figure 6 polymers-15-03364-f006:**
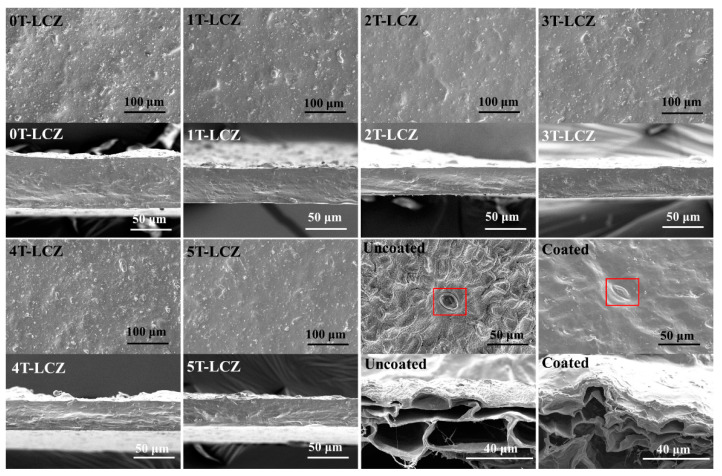
SEM images of the surface and cross-section of the composite coatings, uncoated and coated 3T-LCZ bananas (The red box shows the pores of the banana).

**Figure 7 polymers-15-03364-f007:**
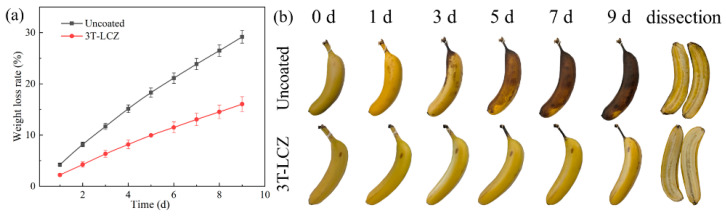
Weight loss rate of bananas during storage (**a**) and changes in the visual appearance of bananas (**b**).

**Table 1 polymers-15-03364-t001:** TS, EB, OP and WVP of the composite coatings.

Samples	TS (MPa)	EB (%)	OP (cm^3^ mm m^−2^ Day^−1^ atm^−1^)	WVP (×10^−11^ g m^−1^ s^−1^ Pa^−1^)
0T-LCZ	31.80 ± 1.36 b	24.80 ± 0.51 ab	0.72 ± 0.13 ab	1.90 ± 0.11 b
1T-LCZ	25.18 ± 2.41 a	25.80 ± 1.33 bc	1.27 ± 0.18 c	2.50 ± 0.02 c
2T-LCZ	29.09 ± 0.95 b	26.76 ± 1.02 c	0.81 ± 0.05 ab	2.01 ± 0.07 b
3T-LCZ	44.57 ± 2.58 d	24.84 ± 1.13 ab	0.66 ± 0.04 a	1.67 ± 0.05 a
4T-LCZ	40.09 ± 2.89 c	23.84 ± 0.39 a	0.83 ± 0.01 ab	1.70 ± 0.05 a
5T-LCZ	39.17 ± 2.02 c	24.84 ± 0.79 ab	0.92 ± 0.03 b	1.71 ± 0.02 a

Different lowercase letters in the same column indicate significant difference (*p* < 0.05).

**Table 2 polymers-15-03364-t002:** Changes in the titratable acid and Vc contents of bananas after 3 days.

Bananas	Titratable Acid (%)	Vc Contents (mg/100 g)
Fresh	0.57 ± 0.01 c	7.37 ± 0.49 b
Uncoated	0.30 ± 0.01 a	4.77 ± 0.19 a
3T-LCZ	0.44 ± 0.01 b	5.31 ± 0.21 a

Different lowercase letters in the same column indicate significant difference (*p* < 0.05).

## Data Availability

The data presented in this study are available on request from the corresponding author.

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
