# Peer review of "Incorporation of Bayberry Tannin into a Locust Bean Gum/Carboxycellulose Nanocrystals/ZnO Coating: Properties and Its Application in Banana Preservation"

_polymers, 2023, doi:10.3390/polym15163364_

Round 1

Reviewer 1 Report

Please, see attached file.

Reviewer 2 Report

Article written correctly, with correct descriptions, scientific discussion and clearly presented results. Prepared even without editorial errors. It fully deserves to be published.

Reviewer 3 Report

This study developed a promising edible locust bean gum coating enriched with bayberry tannins and Zinc oxide to preserve banana's quality and extend shelf life. The effect of bayberry tannins content (1-5%) was investigated. Overall, the study has been conducted properly, and the manuscript is well written. However, it is still required minor revision to make the manuscript better. Please find my comments/suggestions as follows:

- Avoid using acronyms (Vc, LCZ, etc) in the abstract, unless they have been defined. Besides, there are also undefined acronyms, such as WVP, OP etc.

- The procedure for WVP tests are not clearly described. 

Line 95: Suggest using the acronym BT for the tannin, since it has already been defined earlier. 

Line 99: Why the sample is called coatings instead of films? In my opinion, to avoid confusion, the term "films" is more appropriate. 

Line 118: What is OP?

Line 208: Further discussion is required for the statement "the incorporation of tannins impeded the formation of hydrogen bonds among polymers". How hydrogen bonds affect thermal stability of the composites.

- Table 1: Why the properties (OP & WVP) of the films were not affected linearly with the increase of the tannin content? How significant is the results?

- Line 271: The "illustrated" should be "illustrates". There are more sentences required further polishing. 

- Please justify why 5 wt% ZnO and 1 wt% C-CNC were used and mixed with the LBG. 

This is an interesting study. The obtained findings could be useful for helping the development of fruit preservative coating/film materials. The manuscript can be accepted for publication after minor revision.

There are grammar mistakes here and there in the manuscript. It requires further polishing. 
